health and disease and epidemiology/applied mathematics

infectious disease outbreak, non-pharmaceutical interventions, stochastic model, COVID-19, coronavirus, border restrictions

**Authors for correspondence:**
Rachelle N. Binny
e-mail: binnyr@landcareresearch.co.nz
Michael J. Plank
e-mail: michael.plank@canterbury.ac.nz

# Early intervention is the key to success in COVID-19 control

Rachelle N. Binny[1,4], Michael G. Baker[5,6], Shaun C. Hendy[3,4], Alex James[2,4], Audrey Lustig[1,4], Michael J. Plank[2,4], Kannan M. Ridings[3,4] and Nicholas Steyn[2,3,4]

[1]Manaaki Whenua, Lincoln, New Zealand
[2]School of Mathematics and Statistics, University of Canterbury, Christchurch, New Zealand
[3]Department of Physics, University of Auckland, Auckland, New Zealand
[4]Te Pūnaha Matatini: the Centre for Complex Systems and Networks, Auckland, New Zealand
[5]Department of Public Health, University of Otago, Wellington, New Zealand
[6]Maurice Wilkins Centre for Molecular Biodiscovery, Auckland, New Zealand

RNB, 0000-0002-3433-0417; SCH, 0000-0003-3468-6517;
AJ, 0000-0002-1543-7139; MJP, 0000-0002-7539-3465;
NS, 0000-0001-8904-2941

New Zealand responded to the COVID-19 pandemic with a combination of border restrictions and an Alert Level (AL) system that included strict stay-at-home orders. These interventions were successful in containing an outbreak and ultimately eliminating community transmission of COVID-19 in June 2020. The timing of interventions is crucial to their success. Delaying interventions may reduce their effectiveness and mean that they need to be maintained for a longer period. We use a stochastic branching process model of COVID-19 transmission and control to simulate the epidemic trajectory in New Zealand's March–April 2020 outbreak and the effect of its interventions. We calculate key measures, including the number of reported cases and deaths, and the probability of elimination within a specified time frame. By comparing these measures under alternative timings of interventions, we show that changing the timing of AL4 (the strictest level of restrictions) has a far greater impact than the timing of border measures. Delaying AL4 restrictions results in considerably worse outcomes. Implementing border measures alone, without AL4 restrictions, is insufficient to control the outbreak. We conclude that the early introduction of stay-at-home orders was crucial in reducing the number of cases and deaths, enabling elimination.

# 1. Introduction

An outbreak of COVID-19, a novel zoonotic disease caused by the SARS-CoV-2 virus, was first detected in Wuhan, China in November 2019. The virus spread rapidly to other countries, resulting in a pandemic being declared by the World Health Organization in March 2020. Governmental policy responses to COVID-19 outbreaks have varied widely among countries, in terms of the nature and stringency of policy interventions, how quickly these interventions were implemented (electronic supplementary material, table S2) [1] and their effectiveness at reducing spread of the virus [2–4]. While it is tempting to judge the success of interventions by comparison across jurisdictions, this assessment may be confounded by local context that may influence success, as well as by the fact that policy choices can be driven by the severity of initial outbreaks. Models of disease spread played an important role in the design and timing of interventions, but they can also be used *post hoc*, to evaluate the effectiveness of those interventions. For example, Flaxman *et al.* [2] and Brauner *et al.* [5] fitted models of disease dynamics to case count and death data in different countries to estimate the effect of specific non-pharmaceutical interventions on the transmission rate of COVID-19. Modelling studies have indicated the importance of timely interventions for achieving local elimination and averting resurgent waves of COVID-19 in locations like New Zealand and the Australian state of Victoria [6].

In response to the escalating COVID-19 pandemic and the outbreak that was establishing in New Zealand in March 2020, a number of policy interventions were implemented to mitigate risk at the border and reduce community transmission. These measures are detailed in table 1 and included a system of four alert levels. New Zealand moved to Alert Level (AL) 4 on 25 March, at which time there were 315 reported (confirmed and probable) cases, signalling that the government was taking a decisive COVID-19 response that would eventually become an elimination strategy [7]. During the subsequent seven weeks, stringent AL3 or AL4 restrictions, which included stay-at-home orders (see electronic supplementary material, appendix, table S4 for full list of measures) alongside systems for widespread testing, contact tracing and case isolation, were effective at reducing transmission (effective reproduction number, $R_{eff} = 1.8$ [95% CI 1.44, 1.94] prior to AL4; $R_{eff} = 0.35$ [0.28 0.44 during AL4;4,8]). Daily numbers of new cases declined to zero or one by mid-May and the last case of COVID-19 associated with the March outbreak was reported on 22 May. A descriptive epidemiological study by Jefferies *et al.* [9] provides a comprehensive account of SARS-CoV-2 transmission patterns, testing patterns, demographic features and disease outcomes in New Zealand during this period. On 8 June, after 17 consecutive days with no new reported cases, New Zealand officially declared elimination of community transmission following WHO guidelines [10]. This decision was supported by modelling work estimating the probability of elimination at this time [4,6,11]. Between 22 May and 11 August, the only new cases detected were in international arrivals who were required to spend 14 days in government-managed isolation or quarantine facilities [10].

A comparison of the outcomes of New Zealand's COVID-19 response with predicted outcomes from hypothetical alternative actions is important for evaluating the effectiveness of the interventions made and to help refine future response strategies. In this work, we first model the factual scenario using New Zealand's actual intervention timings, then compare this with counterfactual (alternative 'what if') scenarios where policy interventions were implemented earlier or later than occurred in reality, to assess what impact this could have had. For each scenario, we simulate a model of COVID-19 spread and compare key measures, including the peak in daily reported cases, the cumulative numbers of cases and deaths, and the probability of elimination predicted by the model.

In particular, we assess how important New Zealand's decision to move 'hard and early' was for the successful elimination of community transmission following the March–April outbreak. To this end, we compare scenarios with different timings for the introduction of border measures and AL4 to see how these choices could have affected the size of the outbreak. We do not explicitly consider the duration of interventions, although this will be investigated in future work. Indeed, the likelihood of elimination was one of the factors taken into account in New Zealand government decision-making concerning the duration of ALs [12]. While AL4 was successful in achieving elimination, the benefits of elimination had to be weighed against the negative impacts of stringent stay-at-home measures, for example, job losses, financial insecurity, and disruption to education and economic activity. If careful border management could have avoided the need for a lockdown or reduced its intensity, this approach may have been preferable. For instance, Taiwan's early border closure, travel restrictions and 14-day quarantine for those entering the country meant that Taiwan was able to avoid a mass lockdown until May 2021 [13]. We explore whether introducing border restrictions earlier in New Zealand might have been sufficient to eliminate or reduce transmission from international arrivals to the extent where stringent AL4 restrictions could have been avoided or less restrictive measures been sufficient.

**Table 1.** Dates of implementation for policy interventions during New Zealand's COVID-19 response to the March–April 2020 outbreak. All interventions were implemented at 23.59.

| date implemented | policy intervention |
| --- | --- |
| 15 March | 14-day 'self-isolation' (i.e. home quarantine) for all international arrivals |
| 19 March | border closed except to returning residents and citizens |
| 21 March | Alert Level 2 |
| 23 March | Alert Level 3 |
| 25 March | Alert Level 4 |
| 9 April | mandatory 14-day government-managed quarantine for all international arrivals |
| 27 April | Alert Level 3 |
| 13 May | Alert Level 2 (schools and bars remain closed, gathering size limit 10) |
| 18 May | schools reopen |
| 21 May | bars reopen |
| 25 May | gathering size limit increased to 100 |
| 8 June | Alert Level 1: all restrictions lifted except border measures |

## 2. Methods

We simulated a stochastic model of COVID-19 spread in New Zealand [4,14] under a factual scenario using actual timings for border restrictions, border closure and AL4, and for counterfactual scenarios in which implementation of these interventions were either delayed or started earlier. Case data were obtained from Ministry of Health (MoH), containing arrival dates, symptom onset dates, isolation dates and reporting dates for all international cases arriving in New Zealand between February and June 2020. The model is a branching process that is seeded with internationally imported cases and simulates the numbers of new clinical and subclinical infections that are acquired through local transmission each day (see electronic supplementary material, appendix for full model specification and electronic supplementary material, table S1 for list of model parameters). It accounts for delays from infection to symptom onset, and from symptom onset to date of reporting. We assume that the time between an individual becoming infected and infecting another individual (the generation time) follows a Weibull distribution, with a mean and median of 5 days and standard deviation of 1.9 days [15]. The model incorporates individual heterogeneity in transmission rate (e.g. super-spreaders) but individuals are otherwise assumed to be homogeneous and the population well mixed. Contact tracing was in place throughout the entire outbreak and became the primary mode of case detection around the start of AL4 [9]. For simplicity, we do not explicitly model the contact tracing process here. The model assumes that a proportion of clinical infections are undetected by testing and do not get reported, as described in James *et al.* [14]. It assumes that subclinical infections are not reported, do not self-isolate and are 50% as infectious as clinical infections. Each clinical infection is assumed to have a fixed probability of resulting in fatality of 1.32%, calculated using an infection fatality rate (IFR) (for all infections, including subclinical) of 0.88%; this value was obtained by fitting age-specific COVID-19 IFR estimates from international studies [16] to the age distribution of the New Zealand population from 2018 Census data [17].

We account for three classes of interventions for reducing onward transmission: (i) self-isolation (i.e. home quarantine), simulated as a reduction in an individual's transmission rate relative to their transmission when not isolated; (ii) government-managed isolation and quarantine (MIQ), which we assume is 100% effective at preventing onward transmission; and (iii) population-wide control, modelled as a reduction in transmission rate by a factor $C(t)$ at time $t$, relative to no population-wide control, due to restrictions under each of the four ALs. The relative transmission rate is $C(t) = 1$ in the absence of population-wide control and higher ALs correspond to smaller values of $C(t)$ and an associated reduction in effective reproduction number $R_{eff}$. The reproduction number has a large impact on the probability of extinction in a branching process and is related, via the generation time distribution, to the rate of exponential growth during the early phase of an outbreak [18]. Rather than reducing reproduction number in the branching process, the effect of border closure is to reduce the number of seed cases, i.e. the number of new independent branches.

In each scenario, we kept the duration of each AL the same as actually occurred, i.e. 33 days at AL4 followed by 16 days at AL3. We explored the following scenarios (see table 1):

0. **Border restrictions, border closure and AL4 implemented on actual dates.**
1. **Early AL4.** Border restrictions and closure implemented on actual dates, and start of AL4 implemented 5 days early.
2. **Delayed AL4.** Border restrictions and closure implemented on actual dates, and start of AL4:

   a. delayed by 5 days,
   b. delayed by 10 days,
   c. delayed by 20 days.

3. **Early border restrictions.** Border restrictions 5 days earlier; border closure and AL4 on actual dates.
4. **Delayed border closure.** Border closure delayed by 5 days; border restrictions and AL4 on actual dates.
5. **Change in timing of AL4, border restrictions and closure**:

   a. Border restrictions 5 days early and AL4 5 days early; border closure on actual date.
   b. Border closure and AL4 delayed by 5 days; border restrictions on actual date.

6. **No AL4.** No AL3 or AL4 implemented; border restrictions and closure on actual dates.

Border restrictions, border closure and start of AL4 were all implemented at 23.59 so we start simulating their effects on the day after their implementation date. For Scenarios 0, 1, 2, 3 and 6, the model was seeded with the same number of international cases as were reported in case data, using their actual dates of arrival, symptom onset and reporting. For each of these cases, the date of exposure was estimated backwards from the date of symptom onset. In scenarios where border restrictions were implemented on the actual start date (15 March; Scenarios 0, 1, 2, 4 and 6), the self-isolation dates of international cases were set to the same isolation dates as were actually reported. In all scenarios, prior to 9 April, the modelled effect of self-isolation is to reduce an individual's transmission rate to 65% of their transmission when not isolated (following the assumed value used in a model of COVID-19 transmission dynamics by Davies *et al.* [19]). This reflects some risk of onward transmission for cases self-isolating at home. After 9 April, the model assumes that all international cases are placed in MIQ facilities and do not contribute to local transmission. We also simulated a Poisson-distributed random number of international subclinical infections in proportion to the number of international clinical infections (assuming one-third of all infections are subclinical), with arrival and symptom onset dates that were randomly sampled with replacement from the international case data. We assume that these international subclinical infections are not detected and therefore do not self-isolate, but those arriving after 9 April are placed in MIQ and do not contribute to community transmission.

To simulate border restrictions starting 5 days early (Scenarios 3 and 5a), international cases arriving between the earlier start date and the actual start date (11–15 March, inclusive) were assumed to be self-isolated on their date of arrival. To simulate a 5-day delay to border closure (Scenarios 4 and 5b), we delayed the arrival dates (and associated symptom onset, reporting and isolation dates) of international seed cases arriving after 19 March by 5 days. We then allowed for new international cases arriving over these 5 days (e.g. additional non-residents that may have chosen to travel had the border remained open for longer) by seeding an additional Poisson-distributed random number of international cases from 20 March to 24 March, with an average daily number of seeded cases equal to the actual average daily number of international cases arriving during the week prior to 19 March (33 international cases per day). These additional seeded cases were assumed to self-isolate on arrival and their delays from arrival to symptom onset and arrival to reporting were randomly sampled with replacement from the corresponding delays in the actual international case data (obtained from Ministry of Health). We did not attempt to simulate scenarios with delayed border restrictions or earlier border closure because these would have required additional, more substantial modelling assumptions about isolation dates of international arrivals and about the reduction in the volume of international arrivals resulting from border closure. Model predictions would have been highly sensitive to these major assumptions and, without data available to validate them, this would introduce considerable model uncertainty.

For each scenario, we assessed the following key measures describing the dynamics of a COVID-19 outbreak:

1. The maximum number of daily new reported cases and the date on which this occurred (an indicator for the peak load on contact tracing and healthcare systems).
2. The number of daily new reported cases at the end of AL4.
3. Cumulative number of reported cases and the cumulative number of deaths at the end of the seven-week period of AL3–4 restrictions (i.e. end of AL3).
4. Probability of elimination, $P(elim)$, five weeks after the end of AL3.

Throughout Results, the predicted reported cases are the sum of the actual numbers of international clinical infections (plus additional international clinical infections simulated under Scenarios 4 and 5b) seeded in the model and the simulated domestic clinical infections arising by local transmission, after accounting for the probability of detection and delays from infection to reporting. The first measure is useful for assessing whether the contact tracing or health system capacity would have been exceeded. The second measure indicates the daily incidence of cases after four weeks of the most stringent restrictions under AL4, and at the time when restrictions are eased to AL3 (meaning schools, years 1–10, and Early Childhood Education centres can reopen with limited capacity, and non-essential businesses can reopen premises but cannot physically interact with customers; see electronic supplementary material, appendix, table S4). The third measure quantifies the overall health cost of the outbreak. Given New Zealand's elimination strategy [7], we included the fourth measure to assess the likelihood of achieving elimination of community transmission under the different intervention timings. Here, we define elimination as there being no active cases (we assume a case remains 'active' for 30 days after date of exposure) that could contribute to future community transmission. This definition excludes cases in international arrivals after 9 April 2020. Similar definitions employing 28 days after date of exposure (which is twice the maximum incubation period) have been proposed [20]; 30 days provides some buffer for potential error in an infected individual's inferred exposure date. In scenarios resulting in low probabilities of elimination after AL3 restrictions are eased, there is a higher risk of cases persisting undetected in the community and sparking a new outbreak under the weaker AL2 restrictions. For example, gatherings of up to 100 people are permitted under AL2, increasing the risk of super-spreading events. If a new outbreak did occur then another lockdown may be required and the overall health cost would be even higher than that of our third key measure.

In Scenarios 1–5, the model was simulated from 1 February 2020 until a date five weeks after the end of AL3 (end dates therefore vary between scenarios depending on AL timings). In Scenario 6, simulations were run up to 1 February 2021 to allow sufficient time for the outbreak to run its full course. We performed 5000 realizations of the model and report the average value of each key measure as well as the interval range within which 90% of simulation results were contained (in square brackets throughout). In the model, $P(elim)$ was calculated as the proportion of all model realizations that resulted in elimination, defined as no remaining cases within 30 days of infection on the date five weeks after the end of AL3 (or in Scenario 6 with no AL4/3 restrictions, no remaining cases within 30 days of infection on 18 June 2020, i.e. five weeks after the end of actual AL3).

Simulations were run using previously published best-fit estimates of the reproduction number $R_{eff}$ [4]. Hendy *et al.* [4] used the stochastic branching process under the same assumptions applied here, except for a shorter scale parameter of 3.48 days for the isolation-to-report delay distribution, which has little impact on $R_{eff}$ estimates [21]. They compared the average simulated numbers of reported cases per day with observed reported cases and estimated best-fit $R_{eff}$ values by minimizing the root-mean-square error of square root-transformed data, over a time window from 10 March to 27 April. For the period prior to AL4, $R_{eff}$ was estimated to be 1.8 [95% CI 1.44, 1.94], which reflects the average level of transmission in the period prior to 21 March (when there was a heightened public awareness of the risk of community transmission, as the first domestic cases were detected in New Zealand) and during the 2-day periods in AL2 and AL3. During AL4, $R_{eff}$ was estimated to be 0.35 [0.28, 0.44].

Under the ALs 3, 2 and 1, which followed the lockdown, the daily numbers of new cases were too low to obtain reliable estimates of the effective reproduction number $R_{eff}$ from these data using standard techniques (e.g. see Obadia *et al.* [22] for a review of methods). Instead, we simulated the model for assumed values of $R_{eff} = 0.95$, 1.7 and 2.4, respectively. The estimate of $R_{eff} = 2.4$ at AL1 (minimal restrictions, schools and businesses fully open and no limits on gathering size) is in line with estimates reported in Plank *et al.* [23], for the pre-lockdown period of New Zealand's August–September 2020 outbreak, when community transmission was considered to be eliminated and the country was at AL1. Plank *et al.* [23] obtained these estimates using two independent methods: (i) performing Monte Carlo reconstructions of epidemiological transmission trees using contract tracing data [8], and (ii) fitting the Hendy *et al.* [4] stochastic branching process model to daily reported case

count data. The value of $R_{eff}$ at AL2 would probably be lower than at AL1 but greater than 1 due to relatively high activity levels and contact rates as stay-at-home orders are lifted and public venues, businesses and schools reopen. The estimate of $R_{eff} = 1.7$ at AL2 is in the range of estimated values for the pre-lockdown period of the March–April outbreak given in Plank et al. [23]. The $R_{eff}$ for AL3 was chosen to be less than 1; however, we tested the sensitivity of our results to using a value greater than 1. For the scenario with no stringent AL restrictions (Scenario 6), we simulated the model using $R_{eff} = 1.8$ for the entire period (i.e. the same value as was used in all scenarios for the period prior to AL4). This value is lower than that of AL1, reflecting a likelihood of substantial behaviour change and heightened public awareness of the risks of community transmission, even in the absence of any alert level restrictions.

## 2.1. Sensitivity analyses

We further investigated how varying the length of the delay (in days) until the start of AL4 (cf. the delays chosen in Scenario 2) affected key measures. Similarly, we assessed the effect on the key measures of introducing border restrictions 10 days early (cf. 5 days early in Scenario 3). We also tested the sensitivity of our results to different choices of relative transmission rate $C(t)$ (corresponding to changes in $R_{eff}$) under AL3, in particular for a less effective AL3 ($C(t) = 0.46$; $R_{eff} = 1.1$) and a more effective AL3 ($C(t) = 0.30$; $R_{eff} = 0.7$). Previous studies have considered the sensitivity of the branching process model predictions to variations in model parameters whose values are uncertain or context-specific [4,14,21]. Multiple dependencies exist between certain model parameters and the overall population reproduction number $R_{eff}$, as described in James et al. [14]. There is further dependency between $R_{eff}$ and the probability of elimination $P(elim)$ such that a reduction in $R_{eff}$ corresponds to an increase in $P(elim)$. James et al. [14] compared model outputs for different values of the proportion and relative infectiousness of subclinical infections, the individual heterogeneity in transmission rate, and the mean generation time. Changes in these parameters that resulted in a change in the overall population reproduction number, $R_{eff}$, caused a corresponding change in the outbreak trajectory. However, if $R_{eff}$ was set to a fixed value the model was robust to changes in these parameters. Increasing the individual heterogeneity in transmission rate increases the variation between independent realizations of the model and increases $P(elim)$ [24]. The best-fit $R_{eff}$ estimates for AL4 were also relatively insensitive to changes in model parameters [21].

# 3. Results

## 3.1. Scenario 0

To check that the model could accurately replicate the outbreak, we first simulated our model under a factual scenario with border restrictions, border closure and AL4 implemented on the dates they actually occurred. The predicted dynamics of daily new reported cases were a very good visual match to observed daily case data (figure 1) and predicted key measures showed good agreement with the values that were actually observed (table 2, bold text). After moving into AL4, the model prediction and the actual number of daily new reported cases both levelled off at 70–80 for around one week before case numbers started to decline (figure 1). In actual case data, the maximum of 84 new cases per day was observed at the start of this flat-topped peak, while our model predicted a similar maximum (80 [67, 99] new cases per day) occurring 6 days later. By the end of AL3, the model predicted similar cumulative totals to the 1503 cases and 22 deaths actually reported. Five weeks after AL3 restrictions were relaxed, elimination of community transmission of COVID-19 was achieved in 66% of model simulations, giving $P(elim) = 0.66$ (table 2). As we discuss later, higher estimates of probability of elimination reported when New Zealand declared elimination on 9 June were obtained by making use of additional information from case data on the number of consecutive days with zero new reported cases [4]. In the following counterfactual scenarios with alternative timings of interventions, we use Scenario 0 as a baseline for comparing key measures.

## 3.2. Scenario 1: early AL4

Under a scenario where AL4 was implemented 5 days earlier (1 day after border closure), the model predicts slightly lower values for most key measures than were actually observed: daily new cases

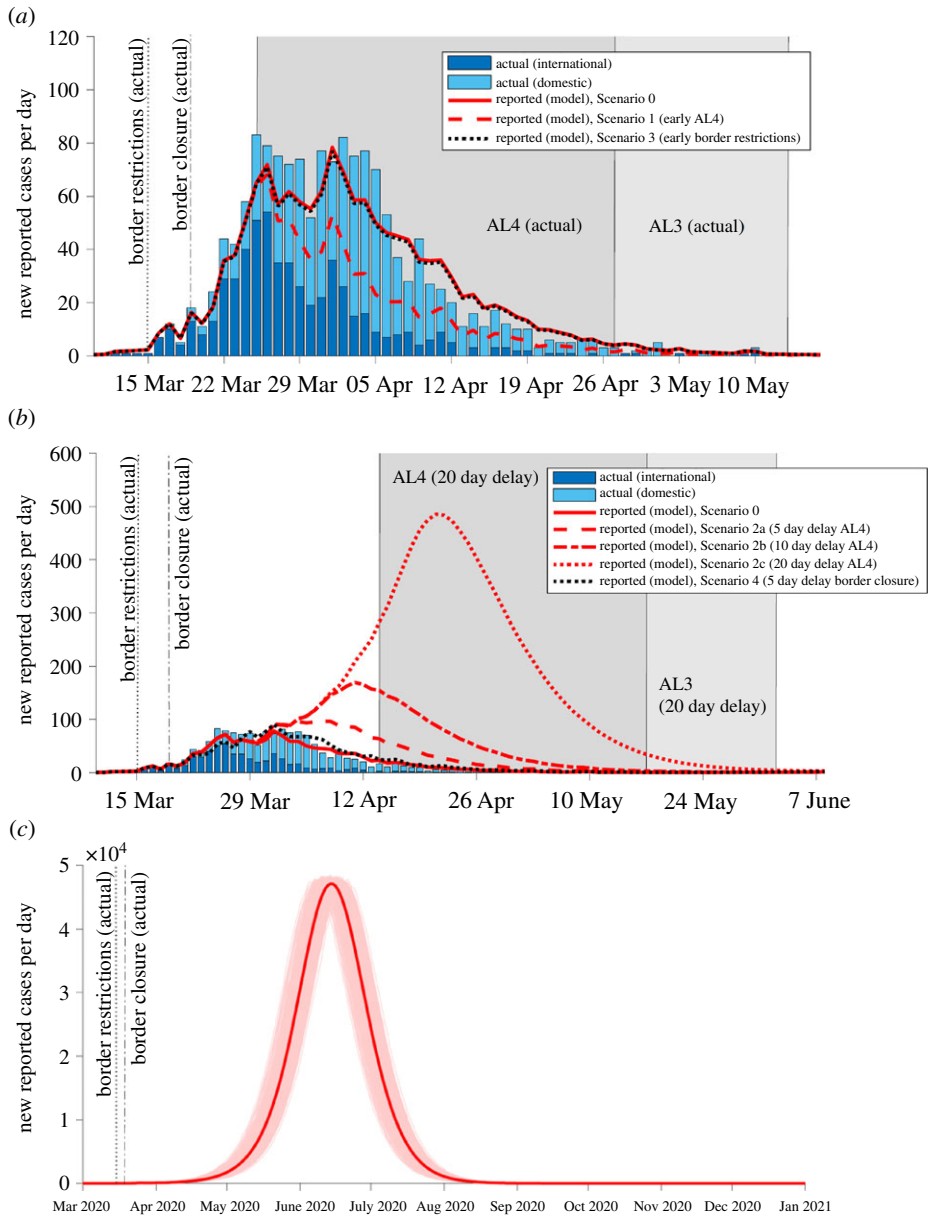

**Figure 1.** Effect of alternative timings of interventions on the trajectory of the outbreak. Number of new reported cases per day predicted by the model (averaged over 5000 simulations) alongside observed reported domestic (light blue bars) and international cases (dark blue bars) (data source: MoH). Model simulated for interventions implemented on their actual start dates and for alternative scenarios with different timings of AL4, border restrictions or border closure (a) Scenario with AL4 started 5 days early (border restrictions and closure on actual start dates) (red dashed; Scenario 1) compared with a scenario where border restrictions were implemented five days early (border closure on actual start date) (black dotted; Scenario 3). (b) Delayed start to AL4 (delays of 5, 10 and 20 days; Scenarios 2a–c; red broken lines) (with border restrictions and closure on actual start dates). Five-day delay to border closure (with border restrictions and AL4 on actual start dates) (black dotted; Scenario 4). (c) No AL4/3 restrictions (border restrictions and closure on actual start dates; Scenario 6) results in an uncontrolled outbreak; faint red lines show the outbreak in individual realizations of the model, the bold red line is the average over all 5000 simulations. Note, x- and y-axis scale differs between figures.

peaked at a lower level of 69 [61, 79] cases around 26 March and at the end of AL4 had dropped to a similar level of four new cases per day as was actually observed (figure 1). By the end of the seven weeks of AL4/3 it predicts approximately 500 fewer cases in total and 10 fewer deaths (table 2; Scenario 1 cf. Scenario 0). However, this estimate should be taken with caution because of the small numbers of daily cases and fine-scale variations involved: for instance, whether an outbreak occurred

**Table 2.** Key measures from alternative scenarios of early or delayed implementation of policy interventions: the maximum number of daily new reported cases, date on which the peak occurs, the maximum number of daily new cases at the end of the simulated Alert Level 4 period, the cumulative number of cases and the total number of deaths at the end of the simulated seven-week period of Alert Level 4/3 restrictions (dates given in 'AL3 ends' and footnotes). For each measure, except P(elim), the mean value from 5000 simulations is reported alongside the interval range, in parentheses, in which 90% of simulations results are contained.

| scenario | border self-isolation | border closed | AL4 starts | AL3 ends | max. new daily cases | date of peak | new daily cases at end of AL4 | cumulative reported cases | total deaths | P(elim) five weeks after end of AL3 |
|---|---|---|---|---|---|---|---|---|---|---|
| **actual** | **15 Mar** | **19 Mar** | **25 Mar** | **13 May** | **84** | **25 Mar** | **3** | **1503** | **22** | — |
| 0 | 15 Mar | 19 Mar | 25 Mar | 13 May | 80 [67, 99] | 31 Mar [26 Mar, 2 Apr] | 4 [1, 8] | 1448 [1208, 1796] | 23 [14, 33] | 0.66 |
| 1 | 15 Mar | 19 Mar | 20 Mar | 8 May | 69 [61, 79] | 26 Mar [25 Mar, 26 Mar] | 4 [1, 7] | 953 [839, 1132] | 14 [8, 21] | 0.63 |
| 2a | 15 Mar | 19 Mar | 30 Mar | 18 May | 108 [84, 139] | 6 Apr [1 Apr, 09 Apr] | 7 [3, 12] | 2373 [1918, 2999] | 39 [26, 55] | 0.57 |
| 2b | 15 Mar | 19 Mar | 4 Apr | 23 May | 179 [137, 233] | 11 Apr [9 Apr, 14 Apr] | 12 [6, 19] | 3988 [3161, 5115] | 67 [48, 91] | 0.38 |
| 2c | 15 Mar | 19 Mar | 14 Apr | 28 May | 503 [382, 661] | 21 Apr [20 Apr, 23 Apr] | 34 [22, 49] | 11 534 [8854, 15 048] | 200 [147, 266] | 0.07 |
| 3 | 10 Mar | 19 Mar | 25 Mar | 13 May | 79 [67, 97] | 31 Mar [26 Mar, 2 Apr] | 4 [1, 8] | 1422 [1194, 1765] | 22 [14, 32] | 0.66 |
| 4 | 15 Mar | 24 Mar | 25 Mar | 13 May | 91 [77, 110] | 1 Apr [31 Mar, 2 Apr] | 5 [1, 9] | 1594 [1359, 1934] | 25 [16, 35] | 0.55 |
| 5a | 10 Mar | 19 Mar | 20 Mar | 8 May | 68 [61, 79] | 26 Mar [25 Mar, 26 Mar] | 4 [1, 7] | 941 [826, 1119] | 14 [8, 21] | 0.63 |
| 5b | 15 Mar | 24 Mar | 30 Mar | 18 May | 120 [97, 152] | 6 Apr [1 Apr, 8 Apr] | 7 [3, 12] | 2501 [2069, 3121] | 41 [28, 56] | 0.53 |
| 6 | 15 Mar | 19 Mar | — | — | 47 592 [47 240, 47 962] | 14 Jun [11 Jun, 17 Jun] | 1127 [841, 1492][a] | 60 443 [45 761, 79 201][b]; 1 812 900 [1 809 600, 1 816 300][c] | 1187 [891, 1565][b]; 31 905 [31 606, 32 204][c] | 0.00[d] |

[a]Evaluated on 27 April 2020 (end of actual AL4); continues to increase after this date.

[b]Evaluated on 13 May 2020 (end of actual AL3); continues to increase after this date.

[c]Evaluated at end of outbreak. Across all realizations, the outbreak had run its full course by approx. October 2020, on average, and the last case reported by 20 December 2020 at the latest.

[d]Evaluated on 18 June 2020 (five weeks after end of actual AL3).

in an aged care facility or not. Five weeks after AL3, the probability of elimination was 63%, slightly lower than in Scenario 0. This counterintuitive result is due to the presence of an international case in the data that had an arrival date prior to the start of AL4 (25 March) but a much later symptom onset date near the end of AL4. In Scenario 0, when international cases are seeded in the model using the actual dates of arrival, symptom onset, reporting and isolation from the case data (see electronic supplementary material, appendix for details), this individual's peak transmission rate occurs during AL4. However, in Scenario 1, the earlier start to AL4 means that the individual is instead most infectious during AL3, where $R_{eff}$ is higher, so this individual infects more people, on average, in this scenario than in Scenario 0. Similarly, any simulated subclinical infections with the same arrival and symptom onset dates (which are sampled with replacement from the international case data) will also be most infectious during AL3. These subclinical infections do not appear in the numbers of reported cases but will reduce the probability of elimination. If this international case outlier is excluded from the data, the model predicts a very similar probability of elimination in both scenarios.

## 3.3. Scenario 2: delayed AL4

Delaying the move into AL4 would have led to a higher peak in daily new cases, and greater cumulative totals of cases and deaths. For a delay of 20 days (Scenario 2c), the outbreak would have reached a considerably higher maximum of close to 500 daily new cases (cf. 80 cases in Scenario 0; table 2, figures 1 and 2). This number would certainly have overwhelmed the contact tracing system, which was already pushed close to capacity in places by the 70–80 daily new cases in late March [25]. After a week in AL4, case numbers would start to decline and by the end of the four weeks in AL4 daily new cases would still have been as high as 34 [22, 49] (close to the actual number of domestic daily reported cases when New Zealand went into AL4 on 25 March). By the end of the seven-week period of stringent restrictions (i.e. end of AL3) the incidence would have dropped to approximately four new cases per day (figure 1), but the cumulative total could have climbed to 11 534 [8854, 15 048] reported cases and 200 [147, 266] deaths, substantially more than Scenario 0 and the 1503 cases and 22 deaths actually reported on 13 May. Additionally, the probability of elimination five weeks after the end of AL3 was only 7%, much lower than in Scenario 0.

## 3.4. Scenario 3: early border restrictions

We next investigated a scenario where border restrictions were put in place 5 days earlier, but border closure and AL4 were started on their actual dates. Border restrictions would therefore have been in place for 9 days (cf. actual 4 days) before the border was closed. Our model predicted this would have had very little impact for the initial trajectory (figure 1) or eventual outbreak size, with values for all key measures similar to those in Scenario 0 (table 2). This suggests that key measures are more sensitive to varying the timing of AL4 than to the timing of border restrictions. In reality, out of the 563 international cases who arrived prior to the start of MIQ and could have contributed to local transmission, only 78 (14%) arrived before border restrictions were implemented on 15 March and were not required to self-isolate. Furthermore, out of these 78 cases, 52 arrived between 10 and 15 March and 19 of these were reported to have voluntarily self-isolated immediately on arrival (the model simulates these 19 cases as being self-isolated on arrival in all scenarios). Therefore, under this scenario, early self-isolation requirements reduce transmissions by a factor $c_{iso} = 0.65$ (electronic supplementary material, appendix, table S1) for only an additional 33 international cases. This reduction is not sufficient to prevent an outbreak, nor does it reduce transmission to an extent where AL4/3 restrictions would not have been necessary to control the outbreak. By contrast, starting AL4 early (Scenario 1) has a stronger effect because it substantially reduces transmissions by a factor $C(t) = 0.15$ (electronic supplementary material, appendix, table S1) for all locally acquired cases (clinical and subclinical) and international cases.

## 3.5. Scenario 4: delayed border closure

Under a scenario where closure of the border (to all except returning residents and citizens) was delayed by 5 days (24 March; 9 days after border restrictions and 1 day before AL4), our model predicted slightly worse outcomes, on average, for key measures compared with Scenario 0. However, due to the stochasticity of individual simulations, the range of key measures always had overlap with the Scenario 0 values and actual values, suggesting a 5-day delay to border closure alone would not have

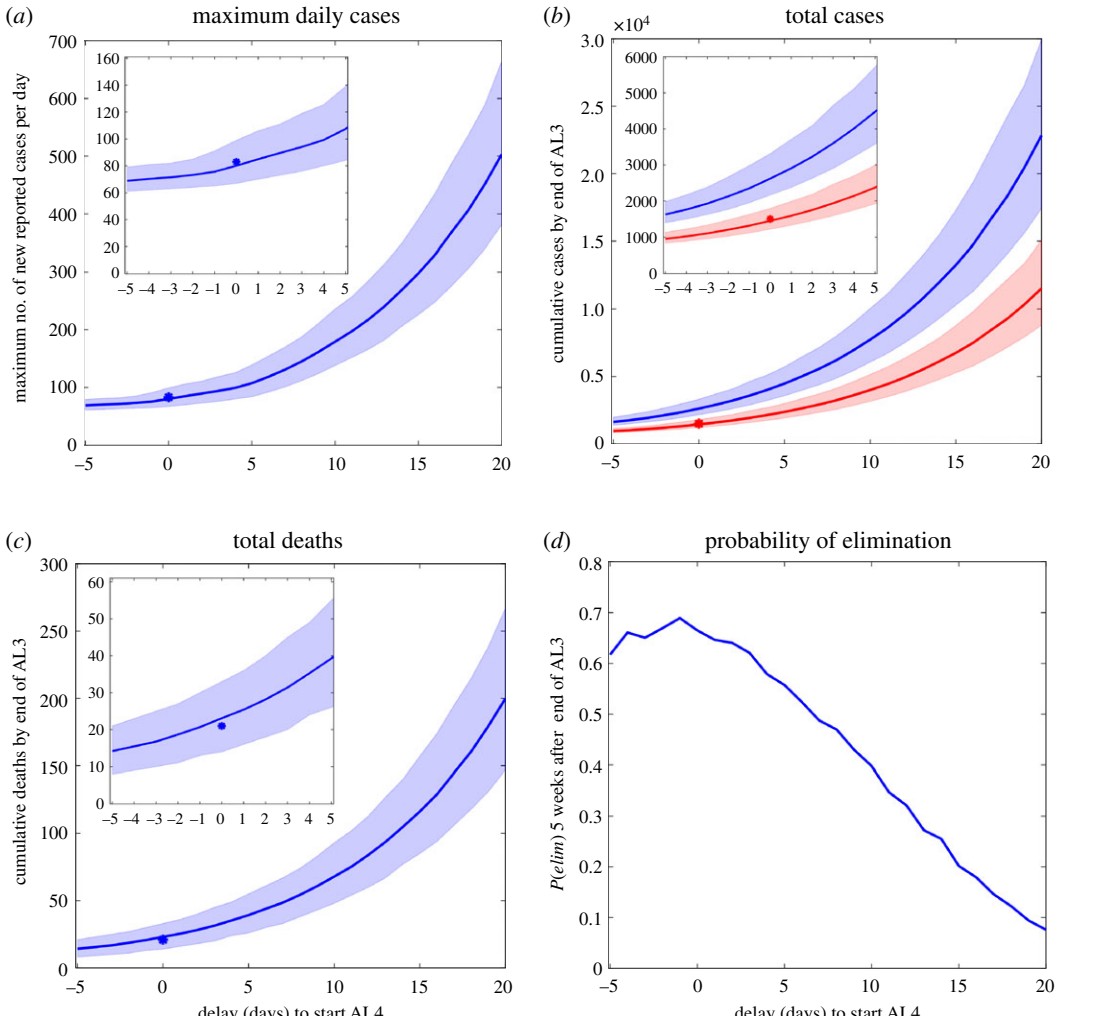

**Figure 2.** Sensitivity of predicted cases and deaths to varying the delay until start of Alert Level 4, up to a maximum delay of 20 days. A negative delay of 5 days represents starting AL4 5 days early (20 March 2020). Border restrictions and closure were implemented on the same dates as actually occurred. (*a*) Maximum number of new reported cases per day predicted by the model (blue line) and actual maximum number of daily reported cases (asterisk). (*b*) Cumulative number of infected individuals (both clinical and sub-clinical) (blue line) and reported cases (red line) predicted by the model and actual number of reported cases (red asterisk). (*c*) Cumulative number of deaths predicted by the model (blue line) and actual number (blue asterisk). (*d*) Probability of elimination, *P(elim)*, five weeks after the end of AL3. Shaded regions indicate the interval range in which 90% of simulation results are contained. Note, *y*-axis scale differs between figures. Insets show close-ups of results for delays from −5 to 5 days.

made a significant difference. A delayed border closure did, however, have a greater impact on the probability of elimination five weeks after AL3 restrictions were relaxed, which was only 55%, compared with 66% in Scenario 0. This reduced probability of elimination is partly due to the additional international clinical infections (captured in the key measures of reported cases) and international subclinicals (not captured in reported cases) arriving prior to the delayed border closure. It is also probably affected by the international case outlier with the pre-MIQ arrival date and late onset date, discussed above.

## 3.6. Scenario 5: change in timing of AL4, border restrictions and closure

After varying the timing of interventions individually, we considered the effects of varying timings for two interventions, to check whether this combination amplifies or counteracts the outcome trends found in previous scenarios. If border restrictions were implemented 5 days early and AL4 came into effect 5 days early (Scenario 5a), this would have led to outcomes very similar to those predicted in

Scenario 1 (where only AL4 started early) (table 2). This again suggests that results are more sensitive to changes in timing for the start of AL4 than to an earlier start to border restrictions.

The outcomes from delaying both border closure and the start of AL4 by 5 days (Scenario 5b) were similar to a 5-day delay to AL4 (Scenario 2a), though slightly worse due to the combined effect of delays to both interventions. Therefore, the timing of AL4 also appears to have a greater impact than timing of border closure. Compared with the factual Scenario 0, daily new cases would have reached a larger maximum of 120 [97, 152] cases on 6 April and by the end of the seven-week period in AL4/3 there would have been close to 1050 more cases in total and nearly 20 more deaths (table 2). The probability of elimination five weeks after AL3 would have also been reduced to 53%.

## 3.7. Scenario 6: no AL4

Finally, we explored the impact of only having border restrictions and border closure in place, but without implementing AL4/3. Under this scenario, the international cases who arrived prior to 9 April and were either in self-isolation or were not isolated have a chance of seeding an outbreak which, without AL4/3 measures to reduce $R_{eff}$ below one, leads to community transmission and a large uncontrolled outbreak. New Zealand would have seen close to 1127 [841, 1492] new cases per day by 27 April, the date on which New Zealand moved from AL4 to AL3 in reality. By 13 May (the date on which New Zealand moved from AL3 to AL2), there could have been over 60 000 cumulative reported cases and over 1100 deaths. New cases would have continued to increase, reaching a peak of 47 592 [47 240, 47 962] daily new cases on 14 June (table 2). By the end of the outbreak, around October 2020 on average, there could have been over 1.81 million reported cases in total and 31 905 [31 606, 32 204] deaths. No simulations resulted in elimination by 18 June (five weeks after end of actual AL3), indicating a 0% chance of COVID-19 having been eliminated by this time, compared with the 66% chance on this date in Scenario 0. This is an important result demonstrating that border measures alone would have been insufficient to prevent a serious outbreak from occurring and that stringent AL4/3 restrictions were necessary to have a chance of eliminating community transmission.

## 3.8. Sensitivity analysis

We assessed the effect that different lengths of delay (in days) until the start of AL4 (figure 2) had on key measures: maximum daily new reported cases; cumulative total reported cases; total infected cases (including both clinical and subclinical); total deaths at end of AL3; and probability of elimination five weeks after the end of AL3. Measures of numbers of cases and deaths increased exponentially with increasing delay to AL4, emphasizing the importance of acting quickly to reduce the risk of large outbreaks arising. Probability of elimination decreased linearly with increasing delays to AL4. Counterintuitively, earlier starts to AL4 slightly reduced the probability of elimination; again, this is caused by the international case outlier discussed previously. If the outlier is excluded from the international case data, the predicted probability of elimination is insensitive to AL4 starting 1 to 5 days early.

Introducing border restrictions 10 days earlier still results in an outbreak and gives very similar results to Scenario 3 (5 days early), with a maximum of 77 [65, 94] new daily cases, 1385 [1166, 1706] cumulative reported cases at the end of AL3, $P(elim) = 0.68$, and other measures the same as in Scenario 3. With border restrictions 10 days earlier, the only difference compared with Scenario 3 is that an additional 16 cases who arrived between 5 and 10 March have their transmission rates reduced (a further two cases arriving in this 5-day period were voluntarily self-isolated on arrival in reality, so are simulated with self-isolation on arrival in all scenarios). This has little impact on the overall contribution to local transmission by all 563 international cases who arrive prior to the start of MIQ, providing further support that border restrictions alone are less effective at reducing community transmission compared with stringent AL restrictions.

We also tested the sensitivity of all key measures to using different values of $R_{eff}$ under AL3 (electronic supplementary material, table S3; $R_{eff} = 1.1$, 0.95 and 0.7). Different choices of AL3 $R_{eff}$ had very little effect on predicted cumulative totals of cases at the end of AL3 and no effect on total deaths at end of AL3. However, the predicted probability of elimination was sensitive to varying AL3 $R_{eff}$; for all scenarios, assuming a lower $R_{eff} = 0.7$ (more effective AL3) gave a $P(elim)$ that was approximately 0.14 higher than with $R_{eff} = 0.95$, while a higher $R_{eff} = 1.1$ (less effective AL3) reduced $P(elim)$ by approximately 0.07. This did not affect our general conclusion that the timing of AL4 has a greater impact on key measures than timing of border measures.

# 4. Discussion

New Zealand's decision to act quickly and to implement stringent restrictions to reduce SARS-CoV-2 transmission meant that, to date, New Zealand has experienced among the lowest mortality rates during the pandemic reported worldwide [26]. Over the course of the March–April 2020 outbreak, a total of 1504 cases and 22 deaths were reported before elimination was achieved in early June 2020. Our results suggest that the timing of AL4 is a much stronger driver of reductions in daily new cases than timings of border measures. This finding makes sense because the effect of AL4 in the model is to greatly reduce $R_{eff}$ for all cases, domestic and international arrivals, to 0.35, while border restrictions reduce the delay until case isolation of international cases only (i.e. international cases have their transmission rates reduced earlier) and border closure reduces the daily numbers of international cases only. Out of the scenarios we considered, an earlier start to AL4 by 5 days resulted in the greatest reduction in numbers of cases and deaths, with approximately 500 fewer cases in total and 10 fewer deaths. However, in reality, the rapid escalation of the COVID-19 situation in mid-March may have made an earlier start to AL4 impractical and would have allowed less time to prepare for ongoing provision of essential services under AL4.

Introducing border restrictions requiring 14-day self-isolation for international arrivals earlier than 15 March would have been unlikely to have much impact on the trajectory of New Zealand's March–April 2020 outbreak, unless such measures were started prior to the first case on 26 February and used methods that were particularly effective. In mid-March, there was a lower global prevalence of COVID-19 and between 2 and 12 cases arrived at the border each day in the week prior to 15 March. The 563 international cases arriving between 15 March and 9 April were already required to self-isolate; had border restrictions been in place prior to the arrival of New Zealand's first case, this would have required self-isolation for, at most, an additional 56 international cases (22 cases who arrived prior to 15 March self-isolated voluntarily immediately on their arrival). Self-isolation is less stringent than MIQ and relies heavily on public compliance. Without additional safety nets, such as official monitoring and support for people who are self-isolating, there is a greater risk of the virus spreading into the community [27]. Self-isolation restrictions for international arrivals can therefore reduce the frequency of cases leaking into the community [8] but outbreaks are still likely to occur, unless AL restrictions are also in place to require strong community-wide social distancing.

Compared with other countries, New Zealand was very quick to close its border to all except returning citizens and residents (electronic supplementary material, table S2). Delaying border closure by 5 days could have led to a slightly larger outbreak, with 10% more cases compared with no delay, but not as large as if AL4 had been delayed by 5 days (64% more cases). The full effect on local transmission potential of the additional international cases expected under a delayed border closure was partially dampened because international cases arriving after 9 April were still placed in MIQ and assumed not to contribute to community transmission. If the timing of this MIQ policy was also delayed, a larger outbreak may have occurred, but we did not model such a scenario here.

If the start of AL4 had been delayed by 20 days, our results suggest New Zealand could have experienced over 11 500 reported cases and 200 deaths, reducing the chance of elimination to only 7%. As with other severe viral disease, the infection fatality risk for COVID-19 is greater for Māori and Pacific peoples (close to 50% higher for Māori than for non-Māori) [28–30]. In scenarios resulting in significantly higher numbers of COVID-19-related deaths (e.g. Scenario 2c), Māori and Pacific communities would probably have been disproportionately affected; however, a population-structured model would be required to assess this consequence in detail. Delaying AL4 would have also increased the chance of a longer lockdown period. With a 20-day delay to AL4, New Zealand could still have been experiencing close to 35 new reported cases per day at the end of a 33-day period at AL4 (cf. under 10 daily new cases in reality). This may have motivated an extension to the lockdown to allow more time for cases to drop below a safe threshold.

The counterfactual scenario with no AL4/3 restrictions (Scenario 6) had disastrous outcomes, including close to 2 million cases and tens of thousands of deaths. This demonstrates that: 1) under the conditions (e.g. level of pandemic preparedness and global COVID-19 prevalence) particular to New Zealand's March–April 2020 outbreak, border restrictions and border closure alone would not have been sufficient to control the outbreak; and 2) New Zealand's national restrictions in combination with its border management, rapid testing and contact tracing, were effective measures that prevented a considerably larger and more prolonged outbreak (i.e. Scenario 6) from occurring during that period.

We report average values for outbreak dynamics, which is appropriate for evaluating the effect of alternative actions and guiding future decision making. However, individual realizations of the stochastic model can deviate (sometimes widely) from the average behaviour. When case numbers are small, as they were in New Zealand, the predicted dynamics are particularly sensitive to fine-scale variations. If $R_{eff}$ is less than 1, an outbreak will eventually die out, but if interventions are relaxed too soon it is still possible for a small number of cases to spark a resurgence. Conversely, when case numbers are small an outbreak can still die out by chance even when $R_{eff}$ is greater than 1. It is important to account for this stochasticity when weighing the effectiveness and risks of different intervention strategies, for example by considering the probability of elimination. On 18 June, five weeks after AL3 restrictions were relaxed, the estimated probability of elimination was 66% in Scenario 0. In reality, as the outbreak died out and more days with zero new cases were observed, this reduced the likelihood of New Zealand being on an upward stochastic trajectory. New Zealand's estimated probability of elimination on 18 June, given there had been 27 consecutive days with zero new cases reported, was approximately 95%, higher than in Scenario 0 [4]. However, this latter estimate required up-to-date information about recent case numbers. For the other scenarios, we explored, bringing in earlier interventions generally had very little impact on probability of elimination, while delaying border closure or AL4 reduced the chance of elimination.

Definitions of the end of an outbreak and approaches to inferring probability of elimination from incidence data vary between studies. Estimates can be highly sensitive to $R_{eff}$ [31] and to surveillance factors, including the degree of time-varying under-reporting, reporting delays, and the interaction of local and imported cases [32]. Our estimates of $P(elim)$ are sensitive to changes in model parameters that result in changes to $R_{eff}$; however, this does not affect our overall conclusions that timing of AL4 is more important for controlling the outbreak than timing of border measures, and that border measures alone are insufficient.

Our model uses a value of $R_{eff} = 0.35$ during AL4 [4], which is consistent with a later estimate of $R_{eff}$ from reconstructions of the epidemiological tree [8]. This is a relatively low value of $R_{eff}$ compared with other countries who implemented interventions roughly equivalent to AL4 [2,4]. A combination of highly effective social distancing in AL4, fast contact tracing, effective case isolation, and the fact that the outbreak occurred at the end of the Southern Hemisphere summer, probably contributed to this low $R_{eff}$ [33]. Disentangling the effects of different interventions introduced in quick succession is challenging (e.g. [2]). For this reason, we apply a single $R_{eff}$ value for the period prior to AL4, rather than different values for the period prior to 21 March and the 2-day periods under AL2 and AL3.

For scenarios where the peak in daily cases exceeded contact tracing system capacity (e.g. Scenario 2c with a maximum 500 daily new cases), this effect would have probably resulted in longer delays to isolation of cases and a higher $R_{eff}$. We did not attempt to model this potential feedback effect and so our results for scenarios where contact tracing system capacity is exceeded may underestimate the outbreak size. Our model assumed a relatively high and constant proportion (75%) of clinical infections are detected and reported. In reality, this proportion can vary over time as testing and contact tracing policies are revised, or as contact tracing and health systems become overloaded in a large uncontrolled outbreak (e.g. Scenario 6). This makes it difficult to benchmark predicted case numbers against empirical data from outbreaks in other countries, where testing and contact tracing regimes may differ from New Zealand. Infection fatality rates also vary between different countries and over time, for example, fatality rates can decrease as new treatments become available. For simplicity, our model assumed a constant IFR of 0.88% (of all infections), which is within the range of IFR estimates reported for other countries [34]. We did not attempt to directly model the burden of COVID-19 on the healthcare system (e.g. numbers of cases requiring hospitalization or intensive care), or the effects of an overwhelmed healthcare system which could have resulted in considerably higher morbidity and mortality.

The key measures of outbreak dynamics assessed here should be considered alongside other measures of economic, social and health impacts (e.g. job losses, consumer spending, impacts for mental health, rates of domestic violence or disrupted education). Particular attention needs to be given to identifying vulnerable groups who may experience inequitable impacts so that future policies can be tailored to support these groups. At the end of AL3, health benefits (e.g. number of cases and deaths avoided) differed between scenarios. However, because the duration spent under AL4/AL3 was fixed at seven weeks for Scenarios 0–5, the direct economic costs of the different scenarios would have been similar. For this reason, we did not attempt to quantify the trade-off between health benefits and economic costs, for example via disability-adjusted life years (DALYs). After the end of AL3, benefits and costs would differ between scenarios depending on the value of $R_{eff}$ for AL1 and

AL2 and on whether or not elimination was achieved. In scenarios with lower probabilities of elimination, it is more likely that New Zealand would have continued to experience new cases while under AL1 and AL2. Since $R_{eff}$ would almost certainly have been greater than 1 at these ALs, this would probably have led to another outbreak and required a second lockdown with its associated costs. Conversely, scenarios with higher probabilities of elimination mean there is less risk of a second lockdown being required. Future work could consider the costs and benefits of alternative scenarios where the duration of time spent in AL4 and AL3 is dictated by the need to achieve a certain outcome, such as a threshold probability of elimination.

Our results are important for reflecting on the effectiveness of intervention timing in New Zealand's COVID-19 response, relative to alternative scenarios, to help guide future response strategies. Early intervention was critical to the successful control of New Zealand's March–April 2020 outbreak. For modelling future disease outbreaks, epidemiological parameters should be updated to reflect changes in national pandemic preparedness (e.g. improved policy and response plans) and behavioural changes influencing the dynamics of future outbreaks. For instance, the degree of compliance with alert level restrictions in future may differ dramatically from the March–April outbreak, resulting in different values of $R_{eff}$. Further work is needed to explore the social dynamics affecting transmission and the effectiveness of interventions, for instance, whether wearing masks in public spaces becomes more common, or whether more people will choose to work from home or avoid travel if a suspected new outbreak is reported or if government action is perceived to be inadequate.

Data accessibility. Matlab code and data required to generate the results and figures in the paper are provided as part of the supplementary material. The results presented in the paper used a dataset, provided by the New Zealand Ministry of Health, on internationally imported COVID-19 cases. To protect privacy of individual travellers, the dataset provided in the electronic supplementary material has been masked, following the method described in the dataset's metadata. Running the model using the masked dataset generates very similar results to those reported in the paper and does not change the conclusions. The original dataset can be requested by emailing the Ministry of Health Data Services team (data-enquiries@health.govt.nz), as is currently stated on the Ministry of Health's COVID-19 Data Resources webpage (https://www.health.govt.nz/our-work/diseases-and-conditions/covid-19-novel-coronavirus/covid-19-data-and-statistics/covid-19-data-resources).

Authors' contributions. All authors conceived and designed the study. A.J., M.J.P. and S.C.H. led development of the model; R.N.B. adapted the model for scenario analysis, ran simulations and generated figures/tables. All authors participated in interpretation and discussion of results. R.N.B. led the writing of the manuscript with substantial contributions from all authors.

Competing interests. We declare we have no competing interests.

Funding. This work was funded by the Ministry of Business, Innovation and Employment and Te Pūnaha Matatini, a New Zealand Centre of Research Excellence.

Acknowledgements. The authors acknowledge the support of StatsNZ, ESR and the Ministry of Health in supplying data in support of this work. We are grateful to Samik Datta, Nigel French, Markus Luczak-Roesch, Melissa McLeod, Anja Mizdrak, Fraser Morgan, Matthew Parry and anonymous referees for comments on an earlier version of this manuscript.

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
