## [Peer Review File · Royal Society Open Science]

Review History

RSOS-210488.R0 (Original submission)

Review form: Reviewer 1

Is the manuscript scientifically sound in its present form?

Yes

Are the interpretations and conclusions justified by the results?

Yes

Is the language acceptable?

Yes

Do you have any ethical concerns with this paper?

No

Have you any concerns about statistical analyses in this paper?

No

Recommendation?

Accept with minor revision (please list in comments)

Comments to the Author(s)

The authors used stochastic simulations exploring the impact of alternative interventions timing scenarios during the early phase of COVID-19 epidemic in New Zealand in 2020 to determine the effectiveness of interventions used based on the timing of interventions. In general, the article is well-written and easy to understand. The simulations were done using some well-established methods with parameters reasonably chosen from other published studies. It is adequate to model the dynamics in the early phase of the epidemic in New Zealand and produce four metrics mentioned in the article that were used to measure the effectiveness of interventions in New Zealand. Modelling results seem to reflect the actual data well (I have an issue regarding the comparison of modelling results to the actual data in my comment). Some important points regarding the content and analysis in the article have been addressed by the other referees. Here, I will add some minor comments:

1. The introduction section is long. It can be made more concise by summarising the intervention timeline and milestones (described in paragraph 2, L75) in a table (i.e., Table 1) without the need of explaining them in details in the paragraph. That should be enough to provide context of the control efforts made the government of New Zealand in the beginning of the epidemic. The mid-to-end section of paragraph 4 of the introduction, starting from L134, described detailed simulation scenarios that were also explained partly in the method section. Can consider to remove that particular section so the introduction can be more concise and no redundancy of information.
2. Figure 1 shows comparison of modelled reported (clinical) cases and reported cases (total of both domestic and international as the barcharts are stacked). Since the model only simulated local transmission, is it correct to assume that the modelled reported cases represented the sum of reported (actual) international cases (or simulated for the delayed border closure scenario) and modelled clinical cases (which then also affected by reporting probability, $p_{det}=0.75$, and also accounting for delay from infection to onset and testing results)? If this is the case, please clarify further either in the figure caption or the results/methods section to avoid confusion.
3. One of the metrics assessed, measured as the daily new reported cases, is labeled as the maximum contact tracing load. From my understanding, the contact tracing process tracks down people who have been exposed by the virus (<https://www.health.govt.nz/our-work/diseases-and-conditions/covid-19-novel-coronavirus/covid-19-health-advice-public/contact-tracing-covid-19>). Hence, the number of the new reported cases shouldn't be a representative of the maximum contact tracing load (as it should be multiple times of the reported cases depending on how many people were 'exposed' by them). Please consider changing the terminology used throughout the article or explain further on why this doesn't really mean the number of people need to be traced.

Review form: Reviewer 2

Is the manuscript scientifically sound in its present form?

Yes

Are the interpretations and conclusions justified by the results?

Yes

Is the language acceptable?

Yes

Do you have any ethical concerns with this paper?

No

Have you any concerns about statistical analyses in this paper?

No

Recommendation?

Major revision is needed (please make suggestions in comments)

Comments to the Author(s)

The authors present a modelling analysis of the impact of border measures and domestic restrictions in New Zealand in early 2020. This provides some useful insights into how timing of introduction influences the success of such approaches in local elimination of COVID-19, and the relative effect of different measures.

Main comments:

- The model uses previously published estimates of R_{eff} as input, rather than fitting the model to the data, but would it would be useful to have some brief details about how these input values were calculated – are the assumptions about generation time etc. in the R_{eff} estimation equivalent to the assumptions in model in this paper?

- I was surprised that cases plateaued almost as soon as AL4 was introduced, given incubation and reporting delays (Fig 1). Presumably this was because the impact of international measures was showing up by that point, rather than AL4 driving the dynamics? If so, it would be good to show the timing of border restrictions in the figure too.

- The model includes isolation of symptomatic cases, but were contacts traced during the pre-AL4 period too? Or was contact tracing only implemented during AL4 (the text suggests something along these lines: L86 - "...alongside systems for widespread testing, contact tracing and case isolation")?

Other comments:

- L32: 'Taiwan's early border closure, travel restrictions and 14-day quarantine for those entering the country have meant that, to date, Taiwan has avoided a mass lockdown' - this may be worth rewording given restrictions introduced in Taiwan in May 2021 (depending on how lockdown is defined).

- L589: 'Delaying border closure by 5 days could have led to a slightly larger outbreak, but not as large as if AL4 had been delayed by 5 days' – could you be more specific here?

Decision letter (RSOS-210488.R0)

Dear Professor Binny

The Editors assigned to your paper RSOS-210488 "Early intervention is the key to success in COVID-19 control" have now received comments from reviewers and would like you to revise the paper in accordance with the reviewer comments and any comments from the Editors. Please note this decision does not guarantee eventual acceptance.

Please be aware that a number of concerns have been raised by the Editors regarding the subject scope of the paper, in particular that it is more clinically focussed than the journal is generally able to consider. Given this, please ensure that you make clear how the work is relevant to a wider audience (ie non-clinical) in your revision - it will remain at the discretion of the Editors whether to accept the revision.

Please submit your revised manuscript and required files (see below) no later than 21 days from today's (ie 02-Sep-2021) date. Note: the ScholarOne system will 'lock' if submission of the revision is attempted 21 or more days after the deadline. If you do not think you will be able to meet this deadline please contact the editorial office immediately.

Associate Editor Comments to Author:

Two reviewers offer a number of comments that need to be addressed before this paper may be considered further. Please ensure you carefully respond to their comments, and provide a point-by-point response as well as a tracked-changes version of the revision - this will help in assessing your revision.

Reviewer comments to Author:

Reviewer: 1

Comments to the Author(s)

The authors used stochastic simulations exploring the impact of alternative interventions timing scenarios during the early phase of COVID-19 epidemic in New Zealand in 2020 to determine the effectiveness of interventions used based on the timing of interventions. In general, the article is well-written and easy to understand. The simulations were done using some well-established methods with parameters reasonably chosen from other published studies. It is adequate to model the dynamics in the early phase of the epidemic in New Zealand and produce four metrics mentioned in the article that were used to measure the effectiveness of interventions in New Zealand. Modelling results seem to reflect the actual data well (I have an issue regarding the comparison of modelling results to the actual data in my comment). Some important points regarding the content and analysis in the article have been addressed by the other referees. Here, I will add some minor comments:

1. The introduction section is long. It can be made more concise by summarising the intervention timeline and milestones (described in paragraph 2, L75) in a table (i.e., Table 1) without the need of explaining them in details in the paragraph. That should be enough to provide context of the control efforts made the government of New Zealand in the beginning of the epidemic. The mid-to-end section of paragraph 4 of the introduction, starting from L134, described detailed simulation scenarios that were also explained partly in the method section. Can consider to remove that particular section so the introduction can be more concise and no redundancy of information.

2. Figure 1 shows comparison of modelled reported (clinical) cases and reported cases (total of both domestic and international as the barcharts are stacked). Since the model only simulated local transmission, is it correct to assume that the modelled reported cases represented the sum of reported (actual) international cases (or simulated for the delayed border closure scenario) and modelled clinical cases (which then also affected by reporting probability, $p_{det}=0.75$, and also accounting for delay from infection to onset and testing results)? If this is the case, please clarify further either in the figure caption or the results/methods section to avoid confusion.

3. One of the metrics assessed, measured as the daily new reported cases, is labeled as the maximum contact tracing load. From my understanding, the contact tracing process tracks down people who have been exposed by the virus (<https://www.health.govt.nz/our-work/diseases-and-conditions/covid-19-novel-coronavirus/covid-19-health-advice-public/contact-tracing-covid-19>). Hence, the number of the new reported cases shouldn't be a representative of the maximum contact tracing load (as it should be multiple times of the reported cases depending on how many people were 'exposed' by them). Please consider changing the terminology used throughout the article or explain further on why this doesn't really mean the number of people need to be traced.

Reviewer: 2

Comments to the Author(s)

The authors present a modelling analysis of the impact of border measures and domestic restrictions in New Zealand in early 2020. This provides some useful insights into how timing of introduction influences the success of such approaches in local elimination of COVID-19, and the relative effect of different measures.

Main comments:

- The model uses previously published estimates of R_{eff} as input, rather than fitting the model to the data, but would it would be useful to have some brief details about how these input values were calculated – are the assumptions about generation time etc. in the R_{eff} estimation equivalent to the assumptions in model in this paper?

- I was surprised that cases plateaued almost as soon as AL4 was introduced, given incubation and reporting delays (Fig 1). Presumably this was because the impact of international measures was showing up by that point, rather than AL4 driving the dynamics? If so, it would be good to show the timing of border restrictions in the figure too.

- The model includes isolation of symptomatic cases, but were contacts traced during the pre-AL4 period too? Or was contact tracing only implemented during AL4 (the text suggests something along these lines: L86 - "...alongside systems for widespread testing, contact tracing and case isolation")?

Other comments:

- L32: 'Taiwan's early border closure, travel restrictions and 14-day quarantine for those entering the country have meant that, to date, Taiwan has avoided a mass lockdown' - this may be worth rewording given restrictions introduced in Taiwan in May 2021 (depending on how lockdown is defined).

- L589: 'Delaying border closure by 5 days could have led to a slightly larger outbreak, but not as large as if AL4 had been delayed by 5 days' – could you be more specific here?

===PREPARING YOUR MANUSCRIPT===

If you have been asked to revise the written English in your submission as a condition of publication, you must do so, and you are expected to provide evidence that you have received language editing support. The journal would prefer that you use a professional language editing service and provide a certificate of editing, but a signed letter from a colleague who is a native speaker of English is acceptable. Note the journal has arranged a number of discounts for authors

using professional language editing services
(<https://royalsociety.org/journals/authors/benefits/language-editing/>).

===PREPARING YOUR REVISION IN SCHOLARONE===

<https://royalsociety.org/journals/authors/author-guidelines/#supplementary-material> to include a suitable title and informative caption. An example of appropriate titling and captioning may be found at https://figshare.com/articles/Table_S2_from_Is_there_a_trade-

off_between_peak_performance_and_performance_breadth_across_temperatures_for_aerobic_sc
ope_in_teleost_fishes_/3843624.

Author's Response to Decision Letter for (RSOS-210488.R0)

See Appendix A.

RSOS-210488.R1 (Revision)

Review form: Reviewer 2

Is the manuscript scientifically sound in its present form?

Yes

Are the interpretations and conclusions justified by the results?

Yes

Is the language acceptable?

Yes

Do you have any ethical concerns with this paper?

No

Have you any concerns about statistical analyses in this paper?

No

Recommendation?

Accept as is

Comments to the Author(s)

The authors have satisfactorily addressed my comments.

Decision letter (RSOS-210488.R1)

Dear Professor Binny,

It is a pleasure to accept your manuscript entitled "Early intervention is the key to success in COVID-19 control" in its current form for publication in Royal Society Open Science. The comments of the reviewer(s) who reviewed your manuscript are included at the foot of this letter.

COVID-19 rapid publication process:

We are taking steps to expedite the publication of research relevant to the pandemic. If you wish, you can opt to have your paper published as soon as it is ready, rather than waiting for it to be published the scheduled Wednesday.

This means your paper will not be included in the weekly media round-up which the Society sends to journalists ahead of publication. However, it will still appear in the COVID-19 Publishing Collection which journalists will be directed to each week (<https://royalsocietypublishing.org/topic/special-collections/novel-coronavirus-outbreak>).

If you wish to have your paper considered for immediate publication, or to discuss further, please notify openscience_proofs@royalsociety.org and press@royalsociety.org when you respond to this email.

on behalf of Kevin Padian (Subject Editor)
openscience@royalsociety.org

Reviewer comments to Author:

Reviewer: 2

Comments to the Author(s)

The authors have satisfactorily addressed my comments.

Appendix A

Response to referees

We are grateful to the Editors and both referees for sharing their time and expertise to review this manuscript. We have addressed all comments as detailed in bold text below. Line numbers refer to the manuscript file that includes tracked changes.

Editor

Please be aware that a number of concerns have been raised by the Editors regarding the subject scope of the paper, in particular that it is more clinically focussed than the journal is generally able to consider. Given this, please ensure that you make clear how the work is relevant to a wider audience (ie non-clinical) in your revision - it will remain at the discretion of the Editors whether to accept the revision.

Our primary focus is on the mathematical modelling of an epidemic under hypothetical policy intervention scenarios. We believe the paper would be of interest to a wide audience with broad expertise in science and mathematics, particularly given the global relevance of the COVID-19 pandemic. In response to comments by Reviewer 1, we have made the Introduction (and other sections, particularly Discussion) more concise. We hope this places greater emphasis on the aims of the work and its relevance for a non-clinical audience.

Reviewer: 1

Comments to the Author(s)

The authors used stochastic simulations exploring the impact of alternative interventions timing scenarios during the early phase of COVID-19 epidemic in New Zealand in 2020 to determine the effectiveness of interventions used based on the timing of interventions. In general, the article is well-written and easy to understand. The simulations were done using some well-established methods with parameters reasonably chosen from other published studies. It is adequate to model the dynamics in the early phase of the epidemic in New Zealand and produce four metrics mentioned in the article that were used to measure the effectiveness of interventions in New Zealand. Modelling results seem to reflect the actual data well (I have an issue regarding the comparison of modelling results to the actual data in my comment). Some important points regarding the content and analysis in the article have been addressed by the other referees. Here, I will add some minor comments:

1. The introduction section is long. It can be made more concise by summarising the intervention timeline and milestones (described in paragraph 2, L75) in a table (i.e., Table 1) without the need of explaining them in details in the paragraph. That should be enough to provide context of the control efforts made the government of New Zealand in the beginning of the epidemic. The mid-to-end section of paragraph 4 of the introduction, starting from L134, described detailed simulation scenarios that were also explained partly in the method section. Can consider to remove that particular section so the introduction can be more concise and no redundancy of information.

We have removed text (L75-88, 146-155) in both paragraphs in the Introduction, and moved other text (L119-126) to Methods. We also identified text in other sections, particularly the Discussion, that was repetitive – we have deleted/reworded text to make this more concise and have reordered some

paragraphs to improve flow. For example, the paragraph in Discussion concerning the probability of elimination results (L732-764) now comes before the discussion of model parameter assumptions, L661-687. The paragraph (L766-777) starting “Our results are important for...” is now the concluding paragraph.

2. Figure 1 shows comparison of modelled reported (clinical) cases and reported cases (total of both domestic and international as the barcharts are stacked). Since the model only simulated local transmission, is it correct to assume that the modelled reported cases represented the sum of reported (actual) international cases (or simulated for the delayed border closure scenario) and modelled clinical cases (which then also affected by reporting probability, $p_{det}=0.75$, and also accounting for delay from infection to onset and testing results)? If this is the case, please clarify further either in the figure caption or the results/methods section to avoid confusion.

This is correct. We now clarify in Methods (L273-277):

“Throughout Results, the predicted reported cases are the sum of the actual numbers of international clinical cases (plus additional international clinical cases simulated under Scenarios 4 and 5b) seeded in the model and the simulated domestic clinical cases arising by local transmission, after accounting for the probability of detection and delays from infection to reporting.”

3. One of the metrics assessed, measured as the daily new reported cases, is labeled as the maximum contact tracing load. From my understanding, the contact tracing process tracks down people who have been exposed by the virus (<https://www.health.govt.nz/our-work/diseases-and-conditions/covid-19-novel-coronavirus/covid-19-health-advice-public/contact-tracing-covid-19>). Hence, the number of the new reported cases shouldn't be a representative of the maximum contact tracing load (as it should be multiple times of the reported cases depending on how many people were 'exposed' by them). Please consider changing the terminology used throughout the article or explain further on why this doesn't really mean the number of people need to be traced.

This is a fair point – the maximum number of new cases isn't equal to the case load on the contact tracing system, it is merely an indicator of the load. For clarity, we have changed the terminology to “maximum/peak daily reported cases” throughout and now only refer to this measure as being an indicator of contact tracing and healthcare load, e.g. in Line 266-268 “The maximum number of daily new reported cases and the date on which this occurred (an indicator for the peak load on contact tracing and healthcare systems).” We have also updated the terminology in the subplot titles in Figure 2 to “maximum daily cases”, “total cases” and “total deaths” to clearly distinguish between these 3 outcome measures.

Reviewer: 2

Comments to the Author(s)

The authors present a modelling analysis of the impact of border measures and domestic restrictions in New Zealand in early 2020. This provides some useful insights into how timing of introduction influences the success of such approaches in local elimination of COVID-19, and the relative effect of different measures.

Main comments:

- The model uses previously published estimates of R_{eff} as input, rather than fitting the model to the data, but would it would be useful to have some brief details about how these input values were calculated – are the assumptions about generation time etc. in the R_{eff} estimation equivalent to the assumptions in model in this paper?

The previously published R_{eff} estimates were obtained by fitting the model to case data. The assumptions and parameter values were the same as those used in this paper except for the isolation-to-reporting delay distribution. We used a larger scale parameter of 6 for this distribution, cf. 3.48 of Binny et al (2020), however sensitivity analyses in Binny et al (2020) found best-fit R_{eff} estimates were relatively insensitive to increasing this scale parameter.

We now provide an overview of these methods (L313-320):

“Simulations were run using previously published best-fit estimates of the reproduction number R_{eff} (Hendy et al., 2021). Hendy et al. (2021) used the stochastic branching process under the same assumptions applied here, except for a shorter scale parameter of 3.48 days for the isolation-to-report delay distribution, which has little impact on R_{eff} estimates (Binny et al., 2020). They compared the average simulated numbers of reported cases per day to observed reported cases and estimated best-fit R_{eff} values by minimising the root-mean-square error of square root-transformed data, over a time window from 10 March to 27 April.”

- I was surprised that cases plateaued almost as soon as AL4 was introduced, given incubation and reporting delays (Fig 1). Presumably this was because the impact of international measures was showing up by that point, rather than AL4 driving the dynamics? If so, it would be good to show the timing of border restrictions in the figure too.

Yes, it is likely that the impacts of border measures were starting to filter through to reported cases (with lags due to the incubation period and the delay from symptom onset to reporting) at the start of Alert Level 4, contributing to the plateau in daily case numbers. We have updated Figure 1 to show the timing of the border restrictions and border closure.

- The model includes isolation of symptomatic cases, but were contacts traced during the pre-AL4 period too? Or was contact tracing only implemented during AL4 (the text suggests something along these lines: L86 - "...alongside systems for widespread testing, contact tracing and case isolation")?

Contact tracing was in place from the very start of the outbreak (pre-AL4) but initially most clinical cases were detected when they presented to healthcare with symptoms. However, the contact tracing system was enhanced over time and from around the start of AL4, the majority of cases were detected by contact tracing. We now clarify this in Methods (L175-177): “Contact tracing was in place throughout the entire outbreak and became the primary mode of case detection around the start of Alert Level 4 (Jefferies et al., 2020). For simplicity, we do not explicitly model the contact tracing process here.”

Other comments:

- L32: 'Taiwan's early border closure, travel restrictions and 14-day quarantine for those entering the country have meant that, to date, Taiwan has avoided a mass lockdown' - this may be worth rewording given restrictions introduced in Taiwan in May 2021 (depending on how lockdown is defined).

Reworded in L141-142 to "Taiwan's early border closure, travel restrictions and 14-day quarantine for those entering the country meant that Taiwan was able to avoid a mass lockdown until May 2021".

- L589: 'Delaying border closure by 5 days could have led to a slightly larger outbreak, but not as large as if AL4 had been delayed by 5 days' – could you be more specific here?

We have specified how much larger the outbreak could have been in terms of percentage increases (L624-627): "Delaying border closure by 5 days could have led to a slightly larger outbreak, with 10% more cases compared to no delay, but not as large as if AL4 had been delayed by 5 days (64% more cases)."